# Implementing Risk-Stratified Breast Screening in England: An Agenda Setting Meeting

**DOI:** 10.3390/cancers14194636

**Published:** 2022-09-24

**Authors:** Lorna McWilliams, D. Gareth Evans, Katherine Payne, Fiona Harrison, Anthony Howell, Sacha J. Howell, David P. French

**Affiliations:** 1Manchester Centre for Health Psychology, Division of Psychology & Mental Health, School of Health Sciences, Faculty of Biology, Medicine and Health, University of Manchester, Oxford Road, Manchester M13 9PL, UK; 2NIHR Manchester Biomedical Research Centre, Manchester Academic Health Science Centre, Manchester University NHS Foundation Trust, Manchester M13 9WU, UK; 3Genomic Medicine, Division of Evolution and Genomic Sciences, School of Biological Sciences, Faculty of Biology, Medicine and Health, University of Manchester, St Mary’s Hospital, Manchester University NHS Foundation Trust, Oxford Road, Manchester M13 9WL, UK; 4Nightingale & Prevent Breast Cancer Research Unit, Wythenshawe Hospital, Manchester University NHS Foundation Trust, Southmoor Road, Wythenshawe, Manchester M23 9LT, UK; 5Manchester Breast Centre, Manchester Cancer Research Centre, University of Manchester, 55 Wilmslow Road, Manchester M20 4GJ, UK; 6Manchester Centre for Health Economics, School of Health Sciences, Faculty of Biology Medicine and Health, University of Manchester, Oxford Road, Manchester M13 9PL, UK; 7Patient representative; 8Division of Cancer Sciences, School of Medical Sciences, Faculty of Biology, Medicine & Health, University of Manchester, Oxford Road, Manchester M13 9PL, UK

**Keywords:** breast cancer, screening, mammography, risk stratification, health inequalities, epidemiology, cost-effectiveness

## Abstract

**Simple Summary:**

A risk-stratified approach to breast cancer screening has gained international interest as a mechanism to better balance the benefits, harms and costs of screening programmes. The aim of our agenda setting meeting was to identify key uncertainties to resolve in order to implement a risk-stratified breast screening programme within England. Findings from a 5-year programme of research assessing the feasibility of such an approach in England were presented before individual discussion groups and an open plenary session regarding what preparation needs to take place. Key findings highlight the need to further develop risk modelling to ensure equity of access to breast cancer risk assessment and ensure a risk-stratified programme is cost-effective. Other issues highlighted related to capacity and capability of the health services to offer an integrated risk assessment pathway that is accessible. Attendees identified ways in which risk-stratification could be implemented to minimise inequity of access to screening.

**Abstract:**

It is now possible to accurately assess breast cancer risk at routine NHS Breast Screening Programme (NHSBSP) appointments, provide risk feedback and offer risk management strategies to women at higher risk. These strategies include National Institute for Health and Care Excellence (NICE) approved additional breast screening and risk-reducing medication. However, the NHSBSP invites nearly all women three-yearly, regardless of risk. In March 2022, a one-day agenda setting meeting took place in Manchester to discuss the feasibility and desirability of implementation of risk-stratified screening in the NHSBSP. Fifty-eight individuals participated (38 face-to-face, 20 virtual) with relevant expertise from academic, clinical and/or policy-making perspectives. Key findings were presented from the PROCAS2 NIHR programme grant regarding feasibility of risk-stratified screening in the NHSBSP. Participants discussed key uncertainties in seven groups, followed by a plenary session. Discussions were audio-recorded and thematically analysed to produce descriptive themes. Five themes were developed: (i) risk and health economic modelling; (ii) health inequalities and communication with women; (iii); extending screening intervals for low-risk women; (iv) integration with existing NHSBSP; and (v) potential new service models. Most attendees expected some form of risk-stratified breast screening to be implemented in England and collectively identified key issues to be resolved to facilitate this.

## 1. Introduction

Breast cancer is the most common form of cancer in the UK with approximately 56,000 cases reported annually equating to 15% of all cancer cases [1]. Organised programmes, such as the NHS Breast Screening Programme (NHSBSP) in England, improves breast cancer outcomes through detecting breast cancers earlier in women aged 50–70 years, however approximately 6000 cancers are diagnosed each year during the interval between screening invitations [2]. In addition, there are harms of breast screening such as false positives and overdiagnosis [3]. 

Risk stratification has been proposed as a means of improving the ratio of benefits to harms, including costs to the healthcare system, of breast screening for some time [4]. In other cancer screening programmes, such as the Welsh and Australian cervical screening programmes, risk-stratified screening interval changes have recently been introduced based on the presence/absence of the key risk factor, human papilloma virus (HPV) infection [5,6]. Currently, the NHSBSP already uses *some* risk stratification, by only routinely inviting women aged 50–70 years where there is evidence of screening benefits, in addition to very high risk screening for example, due to genetic mutations or family history [7]. The National Institute for Health and Care Excellence (NICE) in the UK recommends that women at high risk (≥30% lifetime risk or ≥8% 10-year risk) of breast cancer be offered additional mammographic breast screening from age 40 to 60 and risk reducing medication, such as tamoxifen. NICE also recommend those at moderate risk (17–29% lifetime risk or 3–7.9% 10-year risk aged 40 years) are screened using mammography from age 40–50 and ‘considered’ for risk reducing medication [8]. Additionally, annual MRI surveillance is recommended depending on age and genetic mutation carrier status. Both very high risk screening and risk reducing medication are generally accessed using an opt-in approach via referral to family history, risk and prevention clinics and arranged by local NHS services in England. Ideally, to better balance the benefits to harms ratio, a risk-stratified breast screening programme would be able to identify women at different risk levels and stratify screening intervals and screening modality according to risk.

To allow risk stratification for breast cancer screening at a population-level, much research has focussed on improving risk prediction in the context of national breast screening programmes [9,10]. Models, such as Tyrer-Cuzick [11] and CanRisk [12], now have good discrimination and calibration using a combination of polygenic risk scores (PRS), mammographic density (MD) and self-report risk factors (related mainly to reproductive and hormonal history) and can identify groups of women at high (≥8%) and low risk (<1.5%) [13,14,15]. These models allow the identification of those most likely to be diagnosed with breast cancer and who would benefit most from more frequent screening, access to additional imaging modalities and risk reducing medication, e.g., as per NICE guidelines [8]. Additionally, evidence suggests weight loss interventions are a beneficial option to manage breast cancer risk [16,17]. The Predicting Risk of Cancer at Screening study (PROCAS1) has shown that many women in England are willing to have their breast cancer risk assessed, and receive feedback, as part of the breast screening process [18]. 

International randomised trials in the US (WISDOM) [19] and Europe (MyPeBS) [20], are investigating the efficacy and cost effectiveness of risk-stratified screening compared with the local standard of care and are expected to report in 2027. In both, women identified at higher risk are offered more frequent breast imaging and those at lower risk, less frequent. A second programme of research (PROCAS2) aimed to demonstrate that breast cancer risk assessment (BC-Predict study) can be offered to women at the time of their NHSBSP screening invite in real time [21]. In BC-Predict, self-report information and MD was combined to provide women with their 10-year risk via letter, approximately 6 weeks post-mammogram. Women were also offered the opportunity to receive a risk appointment with a health professional to discuss their risk and any relevant NICE-guideline recommended risk-reducing strategies. 

Thus, there is a pressing need to resolve any uncertainties around implementation outside of a research setting, should these ongoing trials find that risk stratification is indeed effective in reducing cancer stage at diagnosis, and costs. This would allow the NHSBSP to adequately prepare for successful implementation of risk-stratified screening, addressing potential barriers such as staff training and IT system development. Non-UK countries have been considering how to implement a risk-stratified approach to breast cancer screening, mainly in Europe, Australia and Canada [22,23,24,25,26], whilst others have focused more explicitly on specific aspects such as the potential impact on women [27] or on health insurance [28]. An on-going Canadian project (PERSPECTIVE I&I) focuses on identifying additional genetic components of breast cancer risk to incorporate into risk models in addition to developing a population-level implementation framework [29]. This is important given that findings from interviews with professional stakeholders in the Quebec region of Canada indicate the difficulty in reaching consensus of how best to organize a risk-stratified screening programme [30]. Elsewhere, the Western Australia breast screening programme shows it is possible to routinely collect risk factor information from women and the use of which demonstrates the importance of considering a risk-stratified approach, particularly in relation to cancer detection rates for those at higher risk [31]. 

Recent reviews have aimed to identify key research gaps regarding risk-stratified screening for breast cancer, notably one based on a consensus meeting of experts working on various Europe-wide projects on this topic (ENVISION) [32], and a scoping review of evidence gaps [33]. ENVISION explicitly focussed on identifying research gaps regarding risk-stratified screening. It identified four priority areas: breast cancer sub-type specific risk prediction, preventive strategies, measurement of effectiveness and the need for implementation modelling studies [32]. The scoping literature review highlighted that comprehensive breast cancer risk assessments, e.g., including PRS and MD, are required [33]. The current agenda-setting meeting focussed explicitly on what needs to happen to allow risk-stratified screening to be implemented in the NHSBSP. Given this, we included not only academics with highly diverse disciplinary expertise in screening and risk stratification but also members of the public and practising health professionals from diverse professions involved in local screening, risk assessment and medical prevention services and those involved in policy and national implementation of screening (e.g., members of the UK National Screening Committee; UKNSC). 

The aims of the meeting were therefore to ascertain whether attendees with a significant profile in breast screening, breast screening related research and/or screening policy decision-making (a) think risk-stratified breast screening should be implemented and, (b) identify issues that would need to be resolved before implementation into the NHSBSP could proceed.

## 2. Materials and Methods

### 2.1. Design, Participants and Procedure 

A one-day agenda setting meeting was held (March 2022) in hybrid format in Manchester, England. This formed the final stage of the PROCAS2 National Institute of Health and Care Research (NIHR) Programme Grant evaluating the feasibility of risk-stratified breast cancer screening (BC-Predict) [21] specifically in England. Seventy-three invitations were sent via email. Of these 73, 58 (79%) accepted with 38 attending face-to-face and 20 virtually. Attendees were identified due to their relevant expertise, for example directors of local breast screening services, breast radiologists, epidemiologists, health economists, behavioural scientists, statisticians, or individuals with a national role in the NHSBSP and/or UKNSC. The UKNSC includes members of the public therefore we sought to ensure representation from this group by contacting one such member who was involved in a previous breast cancer risk study. The PROCAS2 research programme also includes a lay member co-investigator who has previously accessed a family history, risk and prevention clinic. Both therefore had relevant knowledge about risk-stratified breast screening but importantly could provide a perspective from women likely to be impacted by such a programme. The meeting agenda and key findings from PROCAS2 were provided as four abstracts via email, one week prior to the meeting. Attendees were asked to read the abstracts in advance and 11 relevant publications were included as optional additional reading.

To begin the meeting, after introductions, four members of the PROCAS2 team (DGE, KP, DPF, LM) presented findings from the different programme work packages. The presentations provided: (i) an overview of PROCAS1, PROCAS2 and the main findings from the BC-Predict study; (ii) economic evaluation of risk-stratified breast screening; (iii) evaluation of the psychological impacts of breast cancer risk assessment provision and, (iv) suggestion of key implementation issues to consider. Following this, in one of seven discussion groups (range 49–57 min; mean 51 min), attendees discussed what needs to happen over the next five years for risk-stratified breast screening to be implemented in England, given that the main findings from MyPeBS and WISDOM are expected within five years. Each group included at least one member of the PROCAS2 research team and was facilitated by a member from the wider breast cancer risk research group that led PROCAS2. Group allocation was pre-determined by one co-author (DPF) based on ensuring representation of the relevant disciplines/roles within each group. For example, each group had at least one member who was a breast imaging professional and one member of the UKNSC. 

To guide discussions, ten potential uncertainties (see Table 1) were provided to each group. These indicative uncertainties were identified by the findings from PROCAS2, recent relevant literature, and informed by a series of discussions amongst authors and other members of the PROCAS2 research team. Attendees were also asked to identify and agree on the key uncertainties for implementation along with any concerns before an open plenary session took place. Key points were noted by the facilitators using an online note application (Padlet). The plenary discussion lasted 80 min and was facilitated by DPF with support from LM and DGE. During the plenary, at least one person from each group highlighted the key issue(s) agreed during the earlier discussions. The meeting ended with a brief verbal summary of the meeting by DGE. 

### 2.2. Analysis

The first author listened to audio recordings of each discussion group and the plenary session whilst making notes in a single document to summarise the key points made. No formal transcription of recordings or coding took place. The notes document was used to produce descriptive themes based on the key topics that attendees discussed in groups and focussed on in the plenary session. All points raised in the meeting were captured within the descriptive themes, an approach which has been previously used in articles outlining agenda setting meetings and workshops [34,35]. The results were initially structured in a separate document (Table 1) using the ten indicative uncertainties as sub-headings and involved iterations of combining these or adding new headings to produce the five descriptive themes. For example, communication needs was not an explicit uncertainty or concern provided in Table 1 but was highlighted by all as an important issue. This analysis was highly descriptive, to represent the discussions held, including differences of opinion, and highlight the main concerns and issues that participants identified as requiring resolution for risk-stratified breast screening to be successfully implemented.

## 3. Results

The attendees represented most disciplines relevant to risk stratified screening (see Table 2) with the following breakdown based on *primary* profession and not including the 7 facilitators: 20 academics, 15 clinicians, 5 screening operations/management (this includes members of UKNSC), and 11 in the cancer charity sector. Twelve attendees were present or former members of the UKNSC, UKNSC adult reference group (UKNSC ARG) and BSP Research Advisory Committee. Note that some individuals were both clinical academics involved in the UKNSC and/or NHSBSP operations. There were no health economists present aside from one co-author (KP).

Overall, attendees agreed that a risk stratified approach to breast cancer screening should be the direction of travel in the UK, pending trial results. Reasons given for this included the anticipated benefits and anticipated lack of harms, as well as this approach already taking place in other programmes such as cervical screening and to a limited extent in breast screening with the very high-risk programme for *BRCA1/2* carriers. 

The discussions captured are presented in relation to the following five descriptive themes: risk and health economic modelling; health inequalities and communication with women; extending screening intervals for low-risk women; integration with the existing NHSBSP and; potential new service models. 

### 3.1. Risk and Health Economic Modelling 

Most attendees indicated that the risk models for breast cancer demonstrate good predictive value for breast cancer diagnosis in the majority of the white European population and the most common type(s) of breast cancer, particularly if they use a combination of genetic, mammographic density and self-report information. On the other hand, some attendees were concerned that it is difficult to define what discrimination is good enough without trials or accurate natural history models showing impact on women screened whilst others felt that discrimination would not be good enough should only one component, for example polygenic risk scores or mammographic density alone, be used to determine screening interval. 

Discussions around accuracy of the risk models as implementable tools centred on requiring adequate knowledge about the natural history of the disease; for example, ability to predict different sub-types of breast cancer, and whether the cancer is diagnosed in pre- or post-menopausal women. Secondly, all groups acknowledged that risk is not static but that there is a lack of evidence demonstrating how risk changes over time. Evidence gaps highlighted included the need for better evidence to decide (a) whether to provide women with short term, long-term or lifetime risk and (b) when risk should be reassessed. Linked with this, a concern raised by all groups related to the lack of validation for risk prediction in minority ethnic groups due to lack of available data and that some such groups; for example black African and black Caribbean women, have poorer breast cancer survival rates. However, some attendees expressed confidence that both issues were relatively tractable, noting that research is ongoing so it should be possible to resolve these issues in the near future. Nevertheless, attendees highlighted that it needs to be possible to access these groups in order to assess their risk and assumptions about development of the disease are continually being updated. 

All discussion groups acknowledged that there is no gold standard yet for objective breast density measurement or which imaging technique is best to use if there is high breast density. This was seen as important as it is likely to be a key component of both the risk assessment and screening higher risk women. Some also emphasised that in addition to future risk, it is necessary to decide what type of risk outcome to base risk-stratification on. This may include short-term (e.g., 5 or 10 years) risk of any type of breast cancer, lifetime risk, risk of developing an interval cancer during allocated screening interval, or risk of breast cancer death. This decision will depend on what factors are used to calculate the risk, for example including a PRS score is more effective for estimating lifetime risk and breast density, shorter-term risk. 

The approach to assigning a risk of breast cancer to each woman will affect the economic impact of rolling out a risk-stratified programme. The cost of the risk prediction tool also needs to take account of the required healthcare professional resources to feedback results in a supportive environment. In addition, attendees discussed how adding genetic data to breast cancer risk models would increase the cost of risk prediction substantially. However, the additional predictive value of including genetic data at a population level is not yet known taking into account the different ethnic backgrounds of women offered breast screening. The current economic model presented at the meeting assumed the use of the Tyrer-Cuzick risk model with Volpara™ breast density measurement to assign breast cancer risk. It was acknowledged that to understand the relative cost effectiveness of adding in genetic data, expansion of the current economic model in a future study is required. The need to understand if, and how, to reduce the number of screens over the lifetime of women at low-risk of breast cancer, to free up resources for women at high-risk was also discussed. Before a risk-stratified programme is rolled out, it will be necessary to understand how to keep the programme cost-neutral by introducing risk-stratified screening.

### 3.2. Health Inequalities and Communication with Women

There was agreement that a risk-stratified approach to breast screening may exacerbate existing health inequalities or create new ones. In order to ensure equality of access to information about risk-stratified screening and minimise adverse impact on informed decision-making, attendees discussed different approaches to ensuring meaningful engagement across communities. This did not only focus on characteristics such as ethnicity but also on other groups of women who may already have pre-existing health conditions, including anxiety. Therefore, it was highlighted by some that proactively identifying these groups requiring additional care prior to the offer of risk assessment would ensure adequate support to engage with them and minimise negative consequences of providing risk information. 

It was agreed that there should be a minimum standard of accessible information to enable informed decision-making along the entire pathway however some groups may require additional ‘doses’, i.e., a greater level of intervention, to achieve the same level of engagement. Discussions concerning this mainly centred on providing personal interactions between the service and women given that this may require targeted approaches for different groups of women. Examples of such approaches included developing awareness raising initiatives with trusted community leaders; ensuring culturally appropriate translation of information, especially technical terms; considering the use of ‘pathway navigators’ or health practitioners and developing clear decision aids in multiple formats. The role of wider media campaigns was also discussed as an essential component of communication with women. This included considering that those with higher risk may be concerned about financial implications (e.g., health insurance). 

Given that there may be access barriers related to technology and written information more generally, it was viewed as important to give women the choice about communication preferences. This related to the option of receiving support (another possible role for pathway navigators) to complete the risk assessment itself if self-report information is required and for making informed decisions (with advice from health professionals) about preventive options such as risk-reducing medication.

### 3.3. Extending Screening Intervals for Low-Risk Women

There was agreement that incorporating risk assessment into breast screening is likely to benefit women at moderate or high risk but concerns were raised about whether it is possible to safely extend screening intervals (beyond the current three-year interval) for women at low risk of breast cancer. Establishing evidence that low-risk women are not at increased likelihood of breast cancer mortality if screening frequency is reduced was viewed as paramount before decision-makers are likely to consider this and both screening professionals and women willing to accept it. Similarly, attendees wanted to know more about whether women would accept a longer screening interval if found to be lower risk. 

Some discussion groups focussed on the possibility of the three-yearly screening interval changing to two-yearly, regardless of an individual woman’s risk. Current thinking around this relates to need for reducing the levels of interval cancers in the current NHSBSP and the criticism that the programme intervals are longer than in any other country. This led to considering a two-yearly interval for women at higher than population level risk whilst women at low-risk would remain on the current interval, to minimise both overdiagnosis and the potential fallout of ‘taking screening away’. 

On the flip side, if women at low-risk attend screening at longer intervals and are less likely to require further assessment, this will have a positive impact on service capacity, helping to maintain the roll out as cost-neutral. Additionally, it was acknowledged that other UK national screening programmes have implemented risk adapted screening such as extended intervals for women with HPV negative cervical screening results. 

### 3.4. Integration with Existing NHSBSP 

Discussions about how to implement a risk-stratified approach within the current NHSBSP focussed on on-going challenges within the service which focussed on technology, pathway linkage and capacity issues. In addition, the groups acknowledged there may be a trade-off between using the best possible risk prediction model and feasibility of implementing it successfully within the NHSBSP.

The current IT system, used to invite women every three-years, was viewed as outdated and inefficient (e.g., lack of data on ethnicity of population and separate databases used to invite high risk women more frequently). However, it was widely acknowledged that this is a top priority within the programme. Although newer, future proof technology is expected to be incorporated within the next five years, there was some scepticism about the timescales for this. It was highlighted that this will allow the flexibility required for a risk adapted programme, as well as other innovations that may take place in the lifetime of an IT upgrade. Although viewed as a complex task, existing service failsafe standards and processes can also be adapted in order to minimise potential error, e.g., ensuring the right women are invited to screening at the right time. However, changes to the programme in general also take a long time such as deciding whether to extend the screening age from 50–70 to 47–73 years [36]. 

There was agreement that it was not yet clear how many women would require additional screening (in whatever format that may be) if identified as higher risk, which is required for infrastructure planning, but that it should be possible to estimate this using economic model-based methods. One attendee suggested this should be considered annually for each local service so that women can be invited for screening in a timely manner and that the service can accommodate it. It was acknowledged that women opt into the NHSBSP and need to also opt in to completing a risk assessment; it was not clear how this offer might negatively affect uptake, which has continued to decline in recent years. Equally, it was recognised that uptake would expand to include uptake of risk assessment and risk-stratification pathway in addition to uptake of mammograms. The current organisation of screening is linked to inviting all eligible women from specific GP practices within geographical areas to attend for screening, usually in mobile screening vans. It may however only be possible to offer risk-stratification at static sites, therefore the current invite process may need to change. However, all discussions about this were underpinned by the need for logistic planning, staff training and concern about lack of workforce overall. This predominantly related to limited numbers of radiographers within the current NHSBSP and difficulties with succession planning for this professional group, many of whom are approaching retirement. In addition, the need for appropriate staff to support the risk assessment process was highlighted. Attendees also recognised that women at higher than population risk will be the most likely group to require additional support from health professionals. However, there was a lack of consensus on which health profession would be best suited for this task, given that NHSBSP staff may not necessarily have the skillset or capacity to have these conversations, not all services have fully integrated family history, risk and prevention clinics and even though women may access primary care as a first port of call, GPs may not have the capacity or capability in terms of appropriate training. 

### 3.5. Potential New Service Models

All groups considered alternative service models in settings at least partly outside the NHSBSP. Given that the ultimate aim is to detect cancer as early as possible in order to avoid death, two key aspects related to whether breast cancer risk should be assessed alongside risk of other diseases and the age at which risk assessment should be conducted.

If the goal is to better inform women about their health, some identified that it may be a better idea to integrate risk assessment across several diseases using an integrated approach to cancer early diagnosis (or not just cancer). This was viewed as best delivered in the primary care setting rather than with an individual screening programme given that initiatives such as health checks at age 40 are offered to women via their GP. In addition, risk-stratified approaches are already currently in use for diabetic eye screening and cardiovascular disease and are now being implemented in cervical cancer screening programmes. Targeted lung health checks have also been piloted by NHS England and are currently under consideration by the UKSNC and polygenic risk scores are being deliberated for use in the bowel screening programme. Linked with primary care, several attendees suggested that women could be asked to provide a saliva sample at the time they have their cervical screening appointment in primary care, in order to identify higher risk women at an earlier age (i.e., from age 25) as at the moment, risk-stratified breast screening would only occur when a woman is first assessed in the NHSBSP. However, groups were in agreement that as primary care are currently only indirectly involved in breast screening, it may be more feasible to initially focus on specific aspects of the pathway such as risk-reducing medication for higher risk women.

Similarly, regardless of how the initial risk assessment is carried out, and given that higher risk women are already offered more frequent screening, there must be some involvement with the NHSBSP programme, especially to identify who to invite and when. In the immediate future, it might be possible to offer a simple screening questionnaire prior to a breast screening appointment (e.g., 6 months earlier) to assess interest in risk assessment and identify those who may have higher risk due to family history before providing all risk assessment questions with the mammogram invite or at the appointment itself. One group also identified the possibility of integration with community diagnostic centres, a new NHS England service in development, which could accommodate additional screening for those at higher risk. Lastly, a second opportunity to integrate breast cancer risk assessment, if using polygenic risk scores, was identified due to the growing interest by the Department of Health in the use of genetic information for health more generally, e.g., Our Future Health.

## 4. Discussion

This agenda setting meeting identified cautious enthusiasm for risk-stratified breast screening, and a willingness to engage with the challenges that this raises. The participants identified conceptual issues related to risk modelling and evidence requirements as well as practical objectives to prepare the NHSBSP and the healthcare system more widely for successful implementation of a risk-stratified national breast screening programme.

### 4.1. Strengths and Weaknesses

The meeting included attendees from most of the relevant stakeholder groups, in particular, those who would be involved in the decision to recommend risk-stratified screening to the UK Government (i.e., UKNSC, UKNSC ARG members) and those responsible for implementing any change within the NHSBSP. Including those involved in healthcare policy decisions has previously been highlighted as an important aspect of preparing for implementation of risk-stratified screening in a Spanish study [23]. The lead investigator of the MyPeBS trial attended the meeting. Similarly, UK charity bodies were represented which are an important link between health and social care and the women likely to be offered risk-stratified breast screening. Additionally, two female public contributors attended who provided a patient perspective regarding risk-stratified breast screening at the meeting. Presenting the initial findings from the PROCAS2 research programme allowed for a focused discussion on implementation issues in the English context based on trialing the feasibility of routine implementation in a healthcare setting rather than hypothetical scenarios. However, although efforts were made to identify and invite experts in health economics and healthcare ethics, there was little representation from these disciplines at the meeting, which may have influenced findings. Representatives of the breast screening programmes in the other UK nations, who may have similar or differing views about how risk-stratified screening could be implemented, were not present at the meeting as PROCAS2 was delivered in the English setting. No formal consensus methods were used to gain agreement at the meeting and although similar to other recent meetings [19], the purpose of this agenda-setting meeting was specifically focused on implementation in England, UK. Relatedly, the purpose of this article was to synthesise a description of attendee views rather than conduct an interpretive analysis; similar methods have been used in the field of genomics [34].

### 4.2. Relationship with Existing Literature

Earlier exercises to identify research gaps (e.g., ENVISION) and reviews also recognised that risk stratification is worthwhile to consider implementing for population-level breast screening and concluded that a combination of risk factors should be used to estimate breast cancer risk [19,20]. Further, our agenda setting meeting highlighted overall agreement that some adaptation to the current NHSBSP to more readily identify women at higher risk of breast cancer should take place. However, we also identified several specific practical issues around how risk-stratified screening could be implemented within the context of NHS England, e.g., workforce planning. This was in contrast to previous meetings to produce consensus, which focussed on identifying key research gaps rather than implementation issues.

There was general agreement that it is indeed now possible to predict risk of breast cancer in women of screening age [37]. Indeed, Australian evidence from routinely collected breast cancer risk information in a breast screening programme indicates the utility of identifying women more likely to be diagnosed both with screen-detected cancers and during the screening interval [31]. Yet, in common with a previous review [20], there was some hesitation around the accuracy of breast cancer risk assessment over time in relation to calibration and discrimination despite long-term follow up data indicating this is possible [37]. Similarly, a major concern with risk prediction related to whether the models accurately assess risk in women in ethnic minority groups. This has been documented recently, in relation to polygenic risk indicating this should not yet be used beyond the predominantly white populations in which they have been validated [38] and that different ethnic groups have different risk profiles [39]. Although there was no consensus about whether it would be feasible to implement polygenic risk testing into a risk-stratified NHSBSP, an evaluation of fifteen international cohorts concluded it is beneficial to include in risk estimation [40] and may be more sensitive for risk discrimination [41]. The idea that multiple disease risk assessment (for example, breast cancer, other cancers, cardiovascular disease and diabetes) could be offered to women, particularly in line with genetic testing developments, was considered at the meeting. Recent evidence suggests that some women may prefer to receive information about risk of multiple diseases rather than breast cancer alone [42] and that this multi-disease risk information could be used to engage women in weight-related behaviour change interventions [17].

It was viewed as important and essential that a risk-stratified NHSBSP should not increase costs of the service. Some existing evidence suggests that a risk-stratified screening programme is likely to be cost-effective in England, but this did not include the use of genetic data [43]. Despite being the most useful tool to identify women of White European origin at high and low risk [15], it is yet to be determined how inclusion of polygenic risk would affect cost-effectiveness. Similarly, the added predictive value of using genetics at an inclusive population level is not yet known and may generate specific acceptability concerns related to insurability, as highlighted by Canadian screening policy decision-makers [28]. Identifying high-risk women is also likely to save the Health Service money as prevention with anastrozole has been found to be cost ‘saving’ to the health service [8] The consideration of available healthcare resources and associated costs to the healthcare system should also be viewed in the context of concerns raised around existing workforce, technology and screening capacity issues as also highlighted in qualitative research with health professionals [44,45,46,47,48]. 

It is unsurprising that there is concern regarding how a risk-stratified programme will be received considering that cancer screening is viewed overwhelmingly positively by the general public in the UK [49]. Despite a 70% national uptake of breast screening in England, women may be less willing to accept extended screening intervals if low risk [50,51]. Previous research has also identified that both health professionals and policy decision-makers are apprehensive regarding the safety and acceptability of extending screening intervals for low-risk women [23,45,52]. However, similar to cervical screening [53,54], it may be acceptable so long as women are provided with appropriate information about the safety of such a change [55]. Furthermore, women are receptive to the idea of breast cancer risk assessment overall as part of screening, particularly to access additional screening if perceived to be higher risk and it appears to have no major psychological harms [56,57,58].

The current findings also highlight a major concern that a risk-stratified screening approach may heighten inequity of access for particular groups of women. This has been highlighted previously by Canadian health professionals [59]. Given that some minority ethnic groups have specific access barriers to the NHSBSP [60], and poorer breast cancer outcomes [61], this is an important component of preparing for implementation although some evidence suggests such groups are positive towards the idea of risk stratification [62]. Similarly, approaches identified to minimise widening inequalities, such as community engagement, have already been shown to be effective in improving the odds of breast screening attendance more generally [63]. 

There was some consideration as to whether primary care could support risk-stratified breast screening, however, a recent review suggests that GPs are unfamiliar with breast cancer risk assessment tools and lack confidence in advising on risk reducing medication [64]. Similar findings are highlighted in a recent survey of Canadian health professionals, especially regarding knowledge of polygenic risk [65]. Despite this, recent evidence showed that although mammographic density was not incorporated into the risk assessment, it was possible to identify women at higher risk, and provide risk management, in English primary care practices [66].

### 4.3. Implications for Practice

There is an immediate need to reduce the disparity in access to appropriate services that *currently* offer risk assessment and provide women with access to additional screening and risk-reducing medication [67]. This is an urgent priority and should be addressed prior to any risk-stratified screening programme being implemented given the likely timescale to achieve this. Likewise, it will be essential to have a greater level of integration between services across primary and secondary care with the NHSBSP, ideally centrally organised such as within integrated care systems [68]. This would also allow a coherent communication strategy that is embedded within the community rather than relying on individual areas or primary care practices to support women. Such an approach should facilitate greater uptake amongst women from ethnic minority and low socio-economic status backgrounds, for example primary care endorsements and direct contact from health professionals has been shown to improve access to UK cancer screening programmes [69].

Risk-stratified cervical screening, which has been recently implemented in Wales, will result in 5-yearly intervals for women testing negative for HPV, and 12-month recall for those who test positive. This shift from 3- to 5-yearly intervals for HPV-negative women age 25–49 has already been implemented in Scotland and is recommended in England by the UKNSC. Therefore, such approaches to cancer screening are already under consideration in the screening committee. The current NHSBSP has strict quality assurance standards which may be difficult to integrate with and monitor should aspects of a risk-stratified screening programme be managed out with their remit. These would also require reconfiguration for a risk-stratified service, especially to ensure there are no incidents where women are allocated to the wrong risk group or invited at an incorrect screening interval. It may therefore be most appropriate, and acceptable, to introduce risk assessment to identify women most likely to receive a breast cancer diagnosis in the first instance rather than simultaneously reducing screening intervals for low-risk women. This may alleviate some of the acceptability concerns raised in previous work with health professionals regarding extending screening intervals [52]. It could also be that the number of women who would reach a low-risk threshold is insufficient for the purposes of reallocating resources to those at greater risk. Relatedly, it may be beneficial to do an initial self-report risk assessment prior to women attending their first mammogram appointment given that younger women also develop the disease.

Several information gathering exercises should be undertaken for each screening site to identify a comprehensive picture of (i) local demographics of women eligible for the current NHSBSP in relation to ethnicity and socioeconomic position, (ii) effective local engagement interventions, and (iii) estimated numbers of women in each risk group category. Similar exercises could take place in the other UK nations, or as a collective approach to implementing risk-stratified breast screening. This is an approach being undertaken in Canada who also consider the socio-ethical and legal issues related to risk-stratified breast screening as part of their implementation research [29]. Given that there are now European guidelines for screening in the context of dense breasts [70], it is important to develop a UK response especially as awareness increases; women may be more likely to request density information at their mammogram. Involving a wider number of professionals, such as health practitioners or primary care nurses, to support women in a risk-stratified breast screening programme may alleviate some of the workforce pressures, e.g., at mammogram appointments. Training is however required to upskill primary care staff and increase confidence in discussing risk-reducing medication with women, given that prescriptions are managed in this setting. Similarly, although not yet recommended by the UKNSC, developments in Artificial Intelligence for breast imaging, i.e., reading of images for breast density assessment is likely to become automated, which could minimise the impact of a risk-stratified programme on capacity for radiologists and advanced practitioners. 

Whichever way the NHSBSP decides to proceed, this would need approval from the UKNSC first but should be closely linked with NHS England which will ultimately be responsible for implementing a risk-stratified programme. If this is introduced into the NHSBSP, it is important to implement in such a way that will facilitate evaluation of effectiveness for example, using a stepped wedge trial design.

### 4.4. Implications for Research

Long-term follow up of existing cohorts (e.g., PROCAS, WISDOM, MyPebs), or reassessment of risk where possible, would determine the predictive value of the risk models over time and develop an evidence base for how risk may change in the course of a woman’s screening period. Similarly, the outcomes of the AgeX trial [36] will be important to consider should it recommend the implementation of an extended age range in the NHSBSP, particularly in relation to service capacity. Research is required to assess whether there should be an additional focus on identifying women younger than breast screening age who are at greater risk of breast cancer, prior to their first invite to the NHSBSP. 

Albeit there is some existing UK evidence [55], there is a need to undertake additional acceptability research related to less frequent screening if low risk, with women who have received a low risk breast cancer estimate. Similarly, research exploring how and where women would like to receive information about risk of breast cancer, and whether this should be combined with risk of other diseases, particularly if polygenic risk is part of risk assessment, is warranted.

The need to consider how best to evaluate breast screening outcomes based on breast density, and how to assess it, has been identified in relation to the Australian national programme [24]. The outcomes of on-going breast imaging studies, such as BRAID [71] will be useful to develop the most appropriate pathway for screening those at higher risk. However, it is an on-going concern that clinical trial participants are not necessarily representative of the broader population [72] and any future breast cancer related research, particularly where AI models are trained on existing data, should specifically ensure that strategies are in place to minimise this. 

## 5. Conclusions

With the expectation that risk-stratified breast screening will become accessible to women in England, the main uncertainties and future research priorities have been identified that should be the focus of preparation for implementation. It will be crucial to ensure this new approach to breast screening does not widen health inequalities. 

## Figures and Tables

**Table 1 cancers-14-04636-t001:** Indicative uncertainties provided for group discussions.

How to Organise Risk-Stratified Screening in the NHS Breast Screening Programme to:-	
1	ensure good fit with existing NHSBSP practices
2	increase reach to women generally
3	ensure any change does not furtherdisadvantage to underserved populations, and if anything, reduces inequalities
4	assess the role of general practice
5	consider the need for a simplified model or try for all predictors
6	avoid putting further pressure on the systems (e.g., IT, radiology)
7	adapt models for ethnic minority women
8	consider what aspects (e.g., invitations, mammographic density transfer) can be automated
9	consider extending screening intervals for very low-risk women
10	consider integrating with other NHS health check/health promotion initiatives

**Table 2 cancers-14-04636-t002:** Breakdown of primary profession of attendees (excluding facilitators).

*n*	Profession
12	Breast radiologist, radiographer or surgeon
10	Epidemiologist, health economist (*n* = 1), ethicist (*n* = 1), imaging scientist (*n* = 1) or statistician
7	Charity sector/cancer funding bodies
7	Family history clinician, medical oncologist, dietician or geneticist
6	Behavioural scientist
4	Screening operations/ management
3	General practitioner
2	Public contributor

## Data Availability

The notes document summarising the group discussions will be made available on request by contacting the corresponding author.

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
