# Peer review of "Implementing Risk-Stratified Breast Screening in England: An Agenda Setting Meeting"

_cancers, 2022, doi:10.3390/cancers14194636_

Round 1
Reviewer 1 Report
The paper extensively describes the outcomes of the discussions. The paper is nicely written. However, since no specific risk-based protocol has been discussed, the paper is very general.
Comments:
Risk-based screening seems to focus on adjusting the screening interval. Were also options to provide different tests to high risk women discussed? The introduction could state more clearly what is meant with risk-based screening.
The experts attending the meeting had various and relevant backgrounds. However, only one person had an health economic background, while health economic was one of the main themes. Also, there was only one ethical expert. Would including more experts with the same expertise have led to different conclusions?
WISDOM and myPebs have some experience now in implementation of risk-based screening. Were they consulted to share their experiences of implementation before the meeting?
The meeting focused on implementation in the UK, however, it could be stated that the organization of breast cancer screening is quite similar in other countries (except for the interval). Therefore, the practical issues defined are similar to what has been published before in other countries. Can you elaborate in the discussion more on what is new in this study?
Minor comment:
The is a typo in line 197: expression
Reviewer 2 Report
This manuscript presents a descriptive analysis of stakeholder discussions at a one-day ‘agenda setting’ meeting about the implementation of risk-stratified breast cancer screening in England, UK.
The paper covers a range of relevant issues and provides an overview of recent literature. I have some queries/suggestions for the authors to consider that I believe would improve the manuscript.
1. Introduction (line 80) “women aged 50-70 years when breast cancer is more common” – Targeting of this age group (as opposed to older women) is more about when there is evidence of screening benefit, rather than when breast cancer is more common.
2. I am curious about the “members of the public” (n=2, I think). Without giving away their identities, it would be good to have a better sense of who these people are, how they were identified, and why they were invited. Are they people with quite specific expertise/experience?
3. How were the attendees divided into the discussion groups? Randomly or purposefully?
4. Results (lines 231-233) “in addition to future risk, it is necessary to decide whether to base risk stratification on current risk of breast cancer, risk of interval cancers and risk of breast cancer death, which may depend on what factors are used to calculate the risk.” I don’t understand this sentence. Please rephrase to try and clarify the meaning.
5. Results (line 347) “the ultimate aim is to detect cancer as early as possible” – Surely the goal is ultimately to reduce breast cancer deaths. Aiming purely to detect cancer as early as possible is a recipe for overdiagnosis.
6. Discussion (line 413) “long-term follow up data indicating otherwise.” Could you please clarify what you mean by otherwise? Data indicating there should be no hesitancy re accuracy?
7. Discussion (lines 431-433) “Despite being the most useful tool to identify women at high and low risk, it is yet to be determined how inclusion of polygenic risk would affect this because the added predictive value of using genetics at an inclusive population level is not known.” Why is it described as the most useful tool if the added predictive value is not known?
8. How does the potential shift to risk-based screening fit in with the potential implementation of the age expansion in the NHSBSP? What is the expected timeline for that?
9. Tables – Just a matter of taste, I guess, but I think I’d find the tables much easier to read if the text were left-justified rather than centred.
10. Reference list – Please ensure authors which are organisations appear correctly instead of as if they are a person with a surname and initials.
